# Curcumin—A Viable Agent for Better Bladder Cancer Treatment

**DOI:** 10.3390/ijms21113761

**Published:** 2020-05-26

**Authors:** Jochen Rutz, Andrea Janicova, Katja Woidacki, Felix K.-H. Chun, Roman A. Blaheta, Borna Relja

**Affiliations:** 1Department of Urology, Goethe-University, 60438 Frankfurt am Main, Germany; Jochen.Rutz@kgu.de (J.R.); Felix.Chun@kgu.de (F.K.-H.C.); 2Department of Radiology and Nuclear Medicine, Experimental Radiology, Otto-von-Guericke University, 39106 Magdeburg, Germany; andrea.janicova@med.ovgu.de (A.J.); katja.woidacki@med.ovgu.de (K.W.); borna.relja@med.ovgu.de (B.R.)

**Keywords:** curcumin, bladder cancer, chemoresistance, chemoprotection, bioavailability

## Abstract

Although the therapeutic armamentarium for bladder cancer has considerably widened in the last few years, severe side effects and the development of resistance hamper long-term treatment success. Thus, patients turn to natural plant products as alternative or complementary therapeutic options. One of these is curcumin, the principal component of *Curcuma longa* that has shown chemopreventive effects in experimental cancer models. Clinical and preclinical studies point to its role as a chemosensitizer, and it has been shown to protect organs from toxicity induced by chemotherapy. These properties indicate that curcumin could hold promise as a candidate for additive cancer treatment. This review evaluates the relevance of curcumin as an integral part of therapy for bladder cancer.

## 1. Introduction

The treatment of advanced cancer includes a variety of strategies, e.g., hormonal therapy, radiotherapy, immunotherapy, and chemotherapy. The new immune checkpoint inhibitors, atezolizumab and pembrolizumab, have meanwhile widened the therapeutic options [1]. However, all these approaches offer no cure for once metastasized cancer, and thus, therapy is often palliative rather than curative. The unsatisfactory response and detrimental outcome are due to the development of specific resistance to the applied therapy, resulting in tumor re-activation and re-growth. Severe side effects further limit the benefit of conventional anti-tumor protocols.

During the past decades, increasing numbers of patients have turned to “non-conventional”, i.e., “alternative” or “complementary” cancer treatment options. Though preclinical and clinical studies document numerous positive effects of such unconventional approaches, detailed information about efficacy regarding the suppression or prevention of cancer is still limited or not available. This review focuses on the use of the herbal compound, curcumin, in treating bladder cancer.

## 2. Bladder Cancer

Bladder cancer (BCa) is the ninth most common cancer worldwide, with 550,000 new cases and 200,000 deaths per year [2]. Approximately 75% of patients with BCa suffer from non-muscle-invasive bladder cancer (NMIBC), while 25% already present with muscle-invasive bladder cancer (MIBC) at the time of diagnosis [3]. In 10%–15% of patients with MIBC metastatic cancer is ascertained [4].

The therapeutic strategy depends on the histological identification of the BCa subtypes. Transurethral resection combined with a follow-up intravesical chemotherapy or immunotherapy (Bacillus Calmette-Guerin; BCG) is the main treatment approach for NMIBC [5]. However, high recurrence and progression to a higher tumor stage characterizes this disease, resulting in unsatisfactory patient outcome with a 5-year recurrence rate ranging from 31% to 78% and a progression rate of up to 45% [6].

Once BCa has invaded the *muscularis propria* of the bladder wall (stage T2a and T2b), perivesical fat (T3a, T3b), adjacent organs (T4a), or the pelvic or abdominal wall (T4b), treatment becomes more challenging. Current clinical guidelines recommend cisplatin-based combination chemotherapies as gemcitabine and cisplatin or methotrexate, vinblastine, doxorubicin and cisplatin (MVAC) [7]. However, overall clinical improvement is limited, with response rates of 15%–20% and an improvement in 5-year overall survival of only 5%–8% since the introduction of neoadjuvant chemotherapy 40 years ago. Nearly 60% of patients are non-responders and suffer from muscle-invasive disease, despite chemotherapy [8]. In addition, acquired resistance to cisplatin may trigger tumor relapse and progression. Poor response to therapies, combined with drug resistance and severe side effects, make the improvement and establishment of novel treatment protocols essential.

## 3. Complementary and Alternative Medicine

The dissatisfaction of cancer patients with conventional treatment and subsequent chemotherapeutic side effects has led to expanding the field of anti-tumor therapies to complementary and alternative medicine (CAM). The main reasons given for using CAM are to actively treat the disease, to boost the immune system, to improve physical health, as well as to reduce symptoms [9].

Although the definition of CAM is still imprecise, it is defined as “medical products and practices that are not part of standard medical care”, whereby complementary medicine is used “along with” and alternative medicine “instead of” standard medical treatment [10]. CAM can be divided into three subgroups: use of natural products (e.g., herbs, vitamins), mind and body practices (e.g., meditation, yoga) and “others” (e.g., traditional healers, Ayurvedic medicine, homeopathy) [11]. Worldwide, about 50% of cancer patients variably apply CAM according to tumor type, socio-economic status, and country [12]. Of all the CAM methods, the consumption of plant-derived compounds is most commonly practiced, with a prevalence of 30%–90% [13,14,15].

Nutrition can undoubtedly influence patient health, and several studies point toward distinct anti-cancer mechanisms of natural compounds. Increasing evidence indicates that both therapeutic response and quality of life can be improved when phytodrugs are combined with a standard anti-tumor regimen [16]. Over 2000 plants have been identified that contribute to different CAM strategies [17]. More specifically, a growing number of herbal molecules have meanwhile been identified that reduce the proliferative and invasive properties of various cancers.

Some phenolic compounds exert potent anti-tumor action and of these, the natural polyphenol, curcumin, is the most common anti-cancer phytochemical used in clinical trials [18]. Perusal of the PubMed Central database and the Google Scholar website has revealed that patients with prostate cancer significantly benefit from curcumin [19]. Further studies highlight the biomedical significance of curcumin in treating colorectal cancer [20]. Recently, Naujokat and McKee identified curcumin as being one of the “Big Five” phytochemicals to target cancer stem cells [21].

## 4. Curcumin

*Curcuma longa* (turmeric) is a rhizomatous perennial plant belonging to the *Zingiberacea* family. Although the use of turmeric dates back nearly 4000 years, turmeric has only become popular in the Western world in the last two decades [22]. In Ayuverdic and Chinese medicine, turmeric was traditionally used for treating digestive, liver, and biliary disorders, wounds, gynecological complaints, rheumatism as well as respiratory conditions such as asthma, allergy, or sinusitis [23]. Even though traditional medicine is not based on scientific findings, investigative activity with turmeric compounds for treating diverse inflammatory disorders and cancer has expanded exponentially. 

To date, at least 235 compounds, primarily phenols and terpenoids, have been identified in *C. longa* [24]. Turmeric rhizome consists of 3%–15% curcuminoids, comprised mainly of curcumin (diferuloymethane), demethoxycurcumin, and bisdemethoxycurcumin in a 7:2:1 proportion [25]. Curcumin, the primary active compound, is a yellow-colored polyphenol, poorly soluble in water, but readily soluble in ethanol, methanol, dimethyl sulfoxide (DMSO), or ethyl acetate [26]. Recently, curcumin has been shown to modulate numerous signaling pathways including cell proliferation, cell survival, apoptosis, and cell death, whereby it displays a high potential for anti-cancer therapy by affecting mutagenesis, oncogene expression, cell cycle regulation, and metastasis [27]. Curcumin has been characterized as a potent histone deacetylase (HDAC) inhibitor, which is important, since epigenetic modification through histone modification is a crucial mechanism in cancer development and progression. To take advantage of the role of histone modification in cancer, several synthetic HDAC inhibitors have been developed and approved for clinical use, all of which however are associated with severe side effects. Therefore, curcumin could serve as a dietary phytochemical supplement with HDAC-inhibitory properties, and studies on healthy human volunteers show that oral curcumin administration does reduce HDAC expression without negative side effects [28]. Soflaei et al. summarize the role of curcumin as an epigenetic agent [29]. 

## 5. Curcumin Blocks Bladder Cancer Growth and Proliferation

One of the most frequently discussed topics with regard to the use of curcumin in preclinical treatment of bladder cancer deals with tumor growth and proliferation. Following curcumin administration, tumor cells accumulate in the G_2_/M phase of the cell cycle, suggesting a growth inhibitory effect [30,31,32,33]. Cell cycle arrest in the T24 bladder cancer cell line exposed to 5–12.5 µM curcumin has been shown to be accompanied by a decrease in the cell cycle regulating protein Cyclin A but not of Cyclin B. Although the Cyclin-dependent kinase (Cdk) inhibitor p21 increases in a dose-dependent manner, both cdk1 and cdk2 remain unchanged, pointing to a particular influence of curcumin on Cyclin A-driven processes [33]. However, this mechanism cannot be generalized, since other cell lines (253JB-V and KU7) exhibit decreased p21 expression, along with an elevated p27 level [34]. p27 increase under curcumin exposure has also been observed in RT4 and T24 cells [30]. Recent data obtained from bladder cancer cell lines demonstrate a significant loss of proliferative activity in RT112, UMUC3, and TCCSUP cells following curcumin exposure; however, this loss is triggered by a different molecular mechanism. Cyclin A and B have been shown to decrease in RT112 but increase in UMUC3 cells. Simultaneously, p27 is down-regulated in RT112 but up-regulated in TCCSUP [35]. Epigenetic modulation by curcumin has been demonstrated in UMUC3 cells with a significant increase of the acetylated histones, H3 and H4 [35]. Since all these cell lines differ in grading and staging, curcumin action in specific cell lines may depend on the differentiation status. 

More than 40% of bladder cancers exhibit constitutive activation of the phosphatidylinositol 3-kinase/protein kinase B/mechanistic target of rapamycin (PI3K/AKT/mTOR) pathway [36,37]. Several reports show that mTOR pathway activation may be closely involved in the tumorigenesis of bladder cancer and predict disease progression and poor survival [38,39]. Tian et al. has demonstrated in a rat bladder carcinogenesis model that curcumin strongly checks tumor development by inhibiting the PI3K/AKT/mTOR signaling pathway. Since curcumin can also reduce the expression of Insulin-like Growth Factor 2 (IGF2), the phosphorylation status of its ligand IGF1-receptor (IGF1-R) and insulin receptor substrate 1 (IRS-1), which transmits signals to PI3K, these investigators assume that curcumin suppresses activation of the IGF1-R/IRS-1 axis [40]. These findings have been confirmed with EJ bladder cancer cells, whereby a mechanistic link between PI3K/AKT and the proto-oncogene c-myc has been postulated [41].

Aside from PI3K/AKT, further pathways should be considered. For example, the administration of 5 µM of curcumin has been shown to attenuate the benzidine-triggered proliferation of T24 bladder cancer cells in an extracellular regulated kinase (ERK)1/2-dependent manner [42]. The down-regulation of ERK1/2-signaling is accompanied by diminished expression of the transcription factor activator protein-1 (AP-1), down-regulation of the cell-cycle related proteins PCNA and cyclin D1, and an increase of p21 [42].

A curcumin-induced anti-proliferative response has also been shown to be connected to the attenuated activity of cyclooxygenases (COX). Protein and mRNA expression of COX-2 (but not COX-1) decrease dose-dependently in vitro, correlating with prostaglandin E2 protein levels [33]. Reduced COX-2 protein expression has also been observed in a murine model with orthotopic MB49 tumor cell implants, with Cyclin D1 decreasing concomitantly [43]. This is of interest, since *Cyclin D1* is a downstream target gene of Krüppel-like factor 5 (KLF5), which in cancer is up-regulated as a part of Wnt signaling in a β-catenin-dependent manner [44]. Curcumin has been shown to lower KLF5 protein levels in 5637 and WH bladder cancer cells, dose- and time-dependently. mRNA expression was not affected, indicating a post-transcriptional regulation mechanism [45]. The KLF5 stabilizing proteins, Yes-associated protein (YAP)/transcriptional coactivator with PDZ-binding motif (TAZ), that protect KLF5 from undergoing proteosomal degradation are less expressed as well, suggesting an involvement of the Hippo pathway in curcumin’s anti-cancer effects [45]. These in vitro results have been confirmed in vivo, as exemplified by significant tumor size reduction following curcumin exposure [45].

The nucleolar phosphoprotein Ki-67 is closely linked to the cell cycle and has been found to increase in bladder tumors nearly 20-fold when compared to normal tissue [46]. Treatment of human T24 and rat AY-27 bladder cancer cell lines with 10 µM of curcumin evokes similar proliferation and cell growth reduction, as can be achieved with a siRNA specifically targeting Ki-67 [47]. However, a distinct discrepancy has been detected in the efficacy of curcumin to reduce Ki-67 mRNA (17%), when compared to that of Ki-67-7 (37%). Therefore, although Ki-67 may play an important part in controlling proliferation pathways, curcumin seems to have only a minor impact on this target. 

Another specific target of curcumin is the trophoblast cell surface antigen 2 (Trop2), which is known to enhance the oncogenic activity of bladder cancer. In T24 and RT4 bladder cancer cell lines, curcumin has been shown to attenuate the Trop2 expression with a subsequent decrease of its downstream target Cyclin E1 and elevate the p27 protein level dose-dependently (0–25 µM) [30]. Moreover, the up-regulation of Aurora A, an oncoprotein regulating centrosomal and microtubule activity and controlling chromosome segregation, is attenuated by curcumin administration, inducing the formation of monopolar spindle, G2/M phase arrest, and cell division blockade [32].

Increasing evidence indicates that curcumin modulates the microRNA (miRNA) expression in bladder cancer. Saini et al. has reported that curcumin (10 µM) up-regulates miRNA-203, miRNA-26b, and miRNA-1826 expression in T24 bladder cancer cells [48]. Notably, an increase of miRNA-203 is triggered epigenetically by hypomethylation of the miRNA-203 promotor. In turn, this decreases Act2 and Src protein levels, which are involved in cell growth, proliferation, survival, and motility [38].

Overexpression of miRNA-7641 in bladder cancer has been well documented, as well as its association with tumor development in terms of increased cell viability, colony formation, and suppression of apoptosis [49]. Tumor development has been shown to be dose-dependently reversed by curcumin, accompanied by a decrease of miRNA-7641 and an increase of its direct target, p16 [49]. Curcumin also decreases the expression of miRNA-1246, thereby increasing the expression of its target, p53, in T24 bladder cancer cells [50]. Based on a systematic review, the expression levels of miRNA-21, 143, 155, 214, and 222 have been shown to be closely associated with a longer overall and progression-free survival of bladder cancer patients [51]. Although effects of curcumin on these miRNAs have not been explored in this tumor entity, further data derived from other tumor types provide evidence that curcumin acts on these structures as well.

## 6. Curcumin Induces Apoptosis

Curcumin-induced apoptosis mainly involves mitochondria-mediated apoptotic signaling in numerous cancer types [52]. Curcumin treatment results in a time-dependent increase of p53 [32] with simultaneous down-regulation of the anti-apoptotic protein Bcl-2. On the other hand, expression of the pro-apoptotic Bcl-2-antagonist of-cell-death (Bad) and Bcl-2 associated X protein (Bax) increases in several human bladder cancer cell lines [31]. The pro-apoptotic activity of curcumin has also been confirmed by flow cytometry, demonstrating enhanced caspase-3/7 activity in bladder cancer cells in response to drug treatment [31,53].

Survivin, a member of the inhibitor of apoptosis (IAP) protein family, prevents apoptosis by lowering caspases-3, -7, and -9 [54] and is significantly decreased in bladder cancer cell lines following curcumin application (40 µM) [31]. Decreased expression of survivin has been traced back to the degradation of the specificity protein (Sp)—Sp1, Sp3 and Sp4—all regulating the basal expression of survivin and diminished by curcumin. Curcumin-induced effects on Bcl-2 and cyclin D1 are also assumed to be caused by Sp diminution [31]. Still, the suppression of nuclear factor ‘kappa-light-chain-enhancer’ of activated B-cells (NF-κB) that occurs in the presence of curcumin also correlates with the loss of survivin [34], indicating that curcumin acts dually on survivin via Sp and NF-κB. Sp-proteins are involved in the expression of vascular endothelial growth factor (VEGF) and VEGF receptor 1 (VEGFR1), pointing toward the antiangiogenic potential of curcumin as well [34].

Meanwhile, apoptotic events observed in vitro have been confirmed in a rat model, where curcumin caused condensation and fragmentation of the nuclei [31], decreased Bcl-2 and survivin, and increased Bax [55]. The up-regulation of caspases and activation of apoptotic pathways occurring in several bladder cancer cell lines [47,53,56,57] do not take place in renal tubular epithelial cells, suggesting specific apoptosis induction in the tumor [57].

Poly (ADP-ribose) polymerase-1 (PARP-1) repairs DNA damage by adding poly (ADP ribose) polymers and, therefore, it participates in diverse physiological and pathological functions. In the context of apoptosis, PARP cleavage is evoked by curcumin and related to apoptotic events in 253JB-V and KU7 bladder cancer cells [34].

## 7. Curcumin Suppresses Metastatic Events

Metastatic progression is modulated by direct and indirect mechanisms including receptor-driven interactions of tumor cells with the vessel wall and extracellular matrix proteins, alterations of cytoskeletal dynamics, and epithelial-to-mesenchymal transition (EMT), with the activation of motility-related signaling. Adhesion-blocking properties of curcumin were first shown by Sindhwani et al., who reported that curcumin inhibits tumor implantation in a murine bladder tumor model [58]. Concerning the molecular mode of action, human Trop2 from the tumor-associated calcium signal transducer gene family has been described to be a target of curcumin in bladder cancer cells, modulating invasion and migration via PI3K/AKT and EMT [30].

Further investigation points to curcumin-driven alterations of integrin adhesion receptors. Curcumin suppresses the chemotaxis of RT112, UMUC3, and TCCSUP cells equally well, albeit due to different mechanisms. Diminished chemotaxis of UMUC3 and RT112 has been shown due to the reduced number of cells binding to vascular endothelium, allowing fewer cells to transmigrate, whereas TCCSUP cells are thought to establish a sticky contact to endothelium, hindering migration [59]. The integrin subtypes α3 and β1 are suppressed in all cell lines by curcumin, but a cell-type specific response is seen with further α integrin members. α2 is predominantly reduced in RT112, α5 in UMUC3 and TCCSUP (but not in RT112), and α6 in UMUC3 and RT112 (but not in TCCSUP). Based on blocking studies, the authors conclude that curcumin may generally slow bladder cancer metastasis (at least in vitro). However, this was regulated by different integrin subtypes in the different cell lines [59]. Since all three cell lines are characterized by different staging and grading, curcumin’s molecular mode of action could also depend on the cellular differentiation status. Although this is speculative, evidence has been provided by others that curcumin also acts on EMT proteins [30,60]. Indeed, curcumin has been shown to attenuate the migration and invasion of T24 and 5637 cells by the β-catenin pathway, and it leads to increased epithelial E-cadherin expression and decreased mesenchymal Vimentin and N-cadherin expression [61]. In line with this, the administration of curcumin to mice (50–100 mg/kg bw) down-regulated Mitogen-Activated Protein Kinase (MAPK)-signaling with subsequent EMT-alterations, i.e., elevated E-cadherin and Zonula occludens-1 (ZO-1) and the reduction of Vimentin and N-cadherin [62]. Suppression of the Wnt/β-catenin pathway has also been noted [63]. AKT-suppression, noted by Zhang et al. [30], must also then be interpreted in the context of EMT reversal [64].

Matrix-metalloproteinases (MMPs) enhance the ability of tumor cells to penetrate cellular barriers [65]. In bladder cancer, levels of MMP-2 and MMP-9 have been shown to be diminished by curcumin with an increase of tissue inhibitor of metalloproteinase-2 (TIMP-2) as the relevant underlying mechanism. This suggests an inhibitive influence of curcumin on metastatic progression via MMP pathways [53,66].

## 8. Curcumin-Triggered Immune Response

The process of tumor growth and progression is controlled by the immune microenvironment. Tumor-infiltrating CD4+ and CD8+ T-cells, Treg cells, as well as innate immune cells, such as dendritic cells, macrophages, and NK cells, are all involved in “immunoediting”, finally leading to an immune escape of the tumor [67]. Since this process is accompanied by increased expression of the immune checkpoint protein programmed cell death ligand 1 (PDL1) on both T-cell infiltrates and tumor cells, PDL1 inhibitors have been developed and approved for cancer treatment [68,69]. Curcumin has also been shown to suppress PDL1 in vivo, increase cytotoxic CD8+ T-cells, and decrease Tregs and myeloid-derived suppressor cells (MDSC) [70]. A study on patients reported that curcumin strengthens the anti-tumor immune response by converting cancer patient-derived Tregs to T helper (Th) 1 cells and enhancing interferon-gamma production [71]. 

Aside from regulating adaptive immunity, curcumin also holds potential to modulate innate immunity. Curcumin not only enhances the susceptibility of tumor cells toward NK-mediated cell destruction [72] but also enhances the cytotoxic effect of NK cells associated with the activation of Stat4 and Stat5 proteins [73]. 

Studies are also underway exploring the integration of curcumin into an immunotherapeutic regimen. Curcumin has been shown to distinctly enhance adoptive T-cell therapy with improved cytotoxicity of antigen-specific CD8+ T-cells in tumor-bearing mice [74]. Combined dimethoxy derivative of curcumin together with a PDL1 antibody has been applied to metastasized bladder cancer-bearing mice. Bisdemethoxycurcumin significantly increased CD8+ T-cell infiltrates and CD8+-driven interferon-gamma release and decreased the number of intratumoral MDSC. Simultaneously, the PDL1 antibody protected CD8+ T-cells from exhaustion and therefore facilitated the secretion of interferon-gamma [75]. An elegant concept has been furnished by Hasanpoor and coworkers. In this study, a PDL1-binding peptide was used for the targeted delivery of curcumin to high PDL1-expressing breast cancer cells. The strategy potently improved the cellular uptake and cytotoxicity of curcumin [76]. The combination of curcumin and an anti-PDL1 antibody was found to exert synergistic anti-tumor activity in an ovarian cancer mouse model, indicating that curcumin may indeed serve as an integrative component of current immunotherapy [77]. Meanwhile, a multi-center, open-label, non-randomized, 3-cohort phase 2 study has enrolled patients with recurrent/refractory cervical carcinoma, endometrial carcinoma, or uterine sarcoma. The treatment consists of PD1 blockade (pembrolizumab) combined with radiation and food supplementation, including curcumin-intake [78]. This study is still underway.

## 9. Curcumin Plus Bacillus Calmette-Guerin (BCG) Intravesical Therapy

BCG intravesical therapy is recommended for high-risk and intermediate-risk NMIBC-patients [3]. One and five year disease-progression rates of 11.4% and 19.8%, respectively, of NMIBC patients receiving BCG after transurethral resection of the bladder, show that improved treatment protocols are urgently required [79]. Intravesical instillation of curcumin with BCG in an orthotopic bladder cancer rat model has resulted in fewer tumors compared to BCG alone or to controls [80]. In good accordance, a study on a syngeneic mouse model demonstrates a stronger suppression of bladder cancer growth with combined BCG and curcumin, compared to single drug application [81].

Immunohistochemical evaluation of tumor tissue has revealed the benefit of the BCG-curcumin combination on a molecular basis, with a maximum decrease of the proliferation markers Ki-67 and cyclin D1, suppression of the angiogenesis relevant molecules CD31 and VEGF, and reduction of the anti-apoptosis biomarkers Bcl-2, Bcl-xL, and survivin. Notably, c-myc is down-regulated in the tumor tissue by BCG plus curcumin. c-myc is not only linked to survival, proliferation, and angiogenesis but is also involved in epigenetic control of EMT. Recent reports document a close cross-communication between c-myc and HDAC [82,83], which is highly relevant from a clinical viewpoint. Nearly 90% of all cancers are thought to be caused by epigenetic modification [84], whereby elevated expression of HDACs correlate with a poor prognosis [85]. With the aim to overcome drug resistance, several HDAC inhibitors have been developed and are in clinical use [86]. Kamat et al. [81] did not investigate the HDAC expression level and histone acetylation in their study. However, up-regulation of tumor necrosis factor (TNF)-related apoptosis-inducing ligand (TRAIL) receptors under curcumin treatment and BCG might be traced back to a c-myc–HDAC interaction [83]. Since curcumin has been identified as a natural HDAC-inhibitor, it may therefore carry high potential in supporting a BCG-based regimen.

## 10. Curcumin Plus Chemotherapy

Cho et al. have shown that gemcitabine resistance of bladder cancer cells can be reversed by simultaneous treatment with curcumin (25 µM), leading to an additive cytotoxic effect and reduction of the tumor’s migratory ability [87]. Mechanistically, curcumin was found to increase the expression of the resistance-associated protein ATP Binding Cassette Subfamily C Member 2 (ABCC2) and cleave PARP, whereas deoxycytidine kinase (DCK), cytoplasmic thymidine kinase (TK) 1 and 2 were down-regulated. Elevation of PARP cleavage was interpreted such that curcumin also induced apoptosis [87]. In fact, curcumin potentiated the apoptotic effects of gemcitabine (as well as of paclitaxel) by up-regulating TRAIL and modulating the NF-κB pathway. Curcumin also suppressed COX-2 and VEGF, both of which are linked to proliferation and angiogenesis [88]. The safety of curcumin has already been acknowledged. Systemic exposure to 200–400 mg of curcumin for 14 consecutive days and repetitive treatment every 3 weeks does not increase the incidence of adverse events in cancer patients receiving a gemcitabine-based chemotherapy [89].

Positive effects of curcumin with cisplatin-based therapy have also been shown in vivo and in vitro. Nude mice bearing 253J-Bv cell xenografts display significant reduction of tumor size 27 days following cisplatin–curcumin combination therapy, whereas no response is seen when curcumin or cisplatin is applied alone [90]. It should be noted that cisplatin was applied as an intraperitoneal injection and curcumin by oral gavage. Experiments on an autochthonous animal model of bladder cancer induced by the instillation of N-methyl-N-nitrosourea reveal distinct beneficial effects of curcumin as an intravesical agent, as demonstrated by histological examination and the induction of apoptosis [31]. Notably, curcumin causes no loss in body weight or food intake, nor does it induce signs of toxicity in the animals, indicating its safety.

The molecular mechanism underlying apoptosis induction, when curcumin is combined with cisplatin, is not fully clear. Analysis of several bladder cancer cell lines has revealed two different pathways. Park and colleagues suggest that curcumin may synergistically potentiate cisplatin-induced apoptosis via reactive oxygen species (ROS)-mediated activation of ERK1/2. Either this occurs via a dependent or independent p53 pathway (according to the p53 status of the bladder cancer cells), finally up-regulating pro-apoptotic Bax and down-regulating anti-apoptotic Bcl-2 and the X-linked inhibitor of apoptosis protein XIAP [90]. Curcumin exerts similar anti-tumor effects in both cisplatin-sensitive and cisplatin-resistant lung cancer cell lines. The long-term application of curcumin does not induce resistance to curcumin itself, further enhancing the compound’s potential. Although ongoing studies are required to explore the detailed mode of action of curcumin in a cisplatin-based regimen, simultaneous use may provide an innovative approach to managing human bladder cancer [91].

The influence of curcumin on doxorubicin resistance has not been analyzed in a bladder cancer cell model. Nevertheless, substantial evidence demonstrates that curcumin improves doxorubicin efficacy in colorectal, lung, and breast cancer cells [92]. A pharmacokinetic study has also revealed that curcumin may enhance the absorption of doxorubicin, presumably by decreasing drug efflux via down-regulation of ATP-binding cassette (ABC) drug transporters [93]. Whether this mode of action also holds true for bladder cancer is not yet clear. 

Aside from acting as a chemosensitizer, curcumin has also been proven to protect organs from chemotherapy-induced toxicity. The most prominent adverse event associated with the use of cisplatin is acute kidney damage, affecting up to 60% of patients, whereas the most common adverse effects in patients receiving doxorubicin-based chemotherapy are cardiac disorders [94]. Curcumin reduces cisplatin-induced nephrotoxicity in mice by decreasing serum and renal TNF-alpha (TNFα) and renal monocyte chemoattractant protein (MCP)-1 concentrations [95]. In vivo experiments with breast cancer demonstrate curcumin-associated renoprotection by reducing inflammatory markers such as interleukin (IL)-6 and IL-8 [96]. Aside from its anti-inflammatory effects, curcumin has also been shown to attenuate cisplatin-induced mitochondrial oxidative damage in kidneys [97]. This effect has been confirmed by others, indicating that curcumin prevents alteration in renal mitochondrial bioenergetics, ultrastructure, redox balance, and dynamics [98]. 

Due to its antioxidant, anti-inflammatory, and anti-apoptotic properties, curcumin also exerts diverse cardioprotective effects. Curcumin significantly ameliorates doxorubicin-induced electrocardiographic changes and increases inotropic and chronotropic response to isoprenaline [99]. On a molecular level, curcumin preserves the normal heart architecture and prevents doxorubicin-induced lipid peroxidation, glutathione depletion, and decreased antioxidant enzyme activity. Curcumin also blocks the activation of cardiac lactate dehydrogenase (LDH) and creatine kinase isoenzyme (CK-MB) and reduces the cardiac troponin I (cTnI) level. Down-regulation of the inflammatory response via suppression of the redox-sensitive transcription factor NF-κB and de-activation of TNFα, IL1β, COX-2, and inducible nitric oxide synthase (iNOS) [100] are also attributed to curcumin. NF-κB has recently been shown to be controlled by HDAC [101]. This is intriguing, since HDAC inhibition by curcumin may be the relevant protective mechanism for the kidneys and heart. Still, this is hypothetical and requires further investigation. Table 1 provides an overview of effects of curcumin on bladder cancer in vitro and in vivo. 

## 11. Dose-Dependent Effects of Curcumin

A publication has recently been presented pointing to dual effects of curcumin. Following curcumin treatment, the viability of renal cell carcinoma cells increased in the presence of 5 μM, did not change at 20 μM, and decreased when exposed to 80 μM curcumin, compared to controls. The authors concluded that low curcumin concentrations may protect cells from autophagic cell death, whereas high concentrations may promote cell death [102]. Curcumin at 5 µM inhibited macrophage cell death by decreasing gamma-radiation-induced reactive oxygen species, whereas 25 µM of curcumin increased the reactive oxygen species production and augmented cell death [103]. Both anti-oxidative and pro-oxidative effects of curcumin have also been reported by others, with low dosed curcumin (1–10 µM) being a suppressor and high dosed curcumin (20 µM) being an enhancer of oxidative stress levels [104]. Curcumin’s action on angiogenic events may also depend on the drug concentration. Indeed, evidence has been provided that curcumin plays a pro-angiogenic role at 1–3 μM but an anti-angiogenic one at 10–30 μM [105]. Dose-dependent effects of curcumin on blood vessel regeneration have also been demonstrated by others [106]. Whether these observations are relevant for bladder cancer treatment is still unclear. However, Alavi et al. recently showed that the combination of 5 μM curcumin with 5-fluorouracil significantly reduces the cytotoxicity of fluorouracil, while 15 μM curcumin increases cytotoxicity [107].

## 12. Side Effects of Curcumin

Dose escalation studies demonstrated few curcumin side effects when administered at high concentrations. Grade IV neutropenia and Grade III diarrhea occurred in patients (14%) with advanced or metastatic breast cancer under docetaxel when curcumin was applied orally at 8000 mg/day for seven consecutive days [108]. Intractable abdominal fullness or pain was reported in patients (29%) with advanced pancreatic cancer under gemcitabine when the curcumin dosage was increased to 8000 mg per os (p.o.) daily [109]. No adverse event was attributed to curcumin at 6000 mg (6 consecutive days, p.o.) in patients with metastatic castration-resistant prostate cancer receiving docetaxel/prednisone [110]. Curcumin in complexed form with phospholipids (2000 mg/day continuously) as a complementary approach in patients with pancreatic cancer (plus gemcitabine) did not lead to toxic events, either [111]. The incidence of adverse events in cancer patients (pancreatic and biliary tract cancer) receiving gemcitabine-based chemotherapy did not increase following treatment with 400 mg of curcumin nanoparticles [89]. Short-term intravenous dosing of liposomal curcumin to healthy humans also appeared to be safe, up to a dose of 120 mg/m^2^. However, a transient red blood cell echinocyte formation and an increase in the mean red blood cellular volume was seen in 4% under 400 mg/m^2^ curcumin [112]. In a similar setting, liposomal curcumin was administered as a weekly intravenous infusion for 8 weeks in patients with metastatic tumors. No dose-limiting toxicity was observed at doses between 100 and 300 mg/m^2^. However, of six patients receiving 300 mg/m^2^, one patient developed hemolysis, and three other patients experienced hemoglobin decreases without signs of hemolysis [113].

## 13. Curcumin Delivery

Numerous in vitro and in vivo studies indicate that curcumin acts as both a chemosensitizer and a chemoprotector. Unfortunately, poor solubility, poor absorption, and rapid metabolism are responsible for its low bioavailability and may account for curcumin lacking remarkable anti-cancer activity in clinical trials. Another obstacle to effectivity is that the major metabolic products of curcumin, curcumin glucuronide and curcumin sulfate, are less potent than the parent compound [114,115]. To overcome this, strategies have been developed to optimize bioavailability of curcumin. Suresh and Srinivasan have demonstrated that the concomitant use of curcumin and piperine, isolated from black pepper, enhances the bioavailability of curcumin, presumably by preventing its rapid glucuronidation [116]. A clinical study has confirmed that the curcuminoid–piperine combination significantly improves oxidative and inflammatory status [117]. Bolat et al. combined curcumin and piperine into emulsome nanoformulations, resulting in a water-soluble and stable drug delivery system that elevated the delivery of curcumin and piperine into the tumor cell [118]. A cationic liposome–PEG (polyethylene glycol)–PEI (polyethylenimine) complex has been used as a curcumin carrier by Lin et al. [119], whereas a curcumin phospholipid complex or polymeric curcumin micelles has been recommended by others to increase curcumin half-life [120,121]. Encapsulation of curcumin conjugated with other chemotherapeutic agents (e.g., gemcitabine) could provide an elegant approach to enhancing the anti-tumor potential of the drugs in a synergistic manner [122]. Another technique has been investigated by Mani et al., who expose curcumin-treated cells to visible light. This procedure distinctly enhances efficacy [59]. Finally, the development of potent curcumin analogues and derivatives may also overcome the limitations of the pure compound [123]. In principle, three parts of the curcumin molecule may serve as targets for structural modification: the aromatic rings, the β-diketone moiety, and the two flanking double bonds. Gao et al. recently demonstrated that pyrimidine-substituted curcumin analogues reverse multidrug resistance [124]. Deletion of the β-diketone moiety as well as mono-carbonyl curcumin analogues have been associated with superior toxicity to cancer cells, compared to the mother compound [125,126]. Since curcumin is only poorly soluble in water, the shortening of a redundant hydrocarbon chain or the introduction of hydrophilic groups offers strategies to increase its water solubility. Modifications of curcumin may include the introduction of halogens, hydroxy, nitro, and methoxy groups [127].

## 14. Conclusions

In view of the proven manifold anti-tumor properties of curcumin on the molecular level, coupled with nontoxicity, this compound could serve as an augmented treatment option for bladder cancer (Figure 1). Studies on humans are sparse; therefore, ongoing trials are mandatory to evaluate whether the results obtained in vitro or in vivo will prove clinically beneficial. Curcumin dosage, bioavailability, the optimal indication, and potential toxic effects are further issues that require attention. When these issues have been clarified, the question of whether curcumin can become an important new therapeutic option in bladder cancer treatment may be answered.

## Figures and Tables

**Figure 1 ijms-21-03761-f001:**
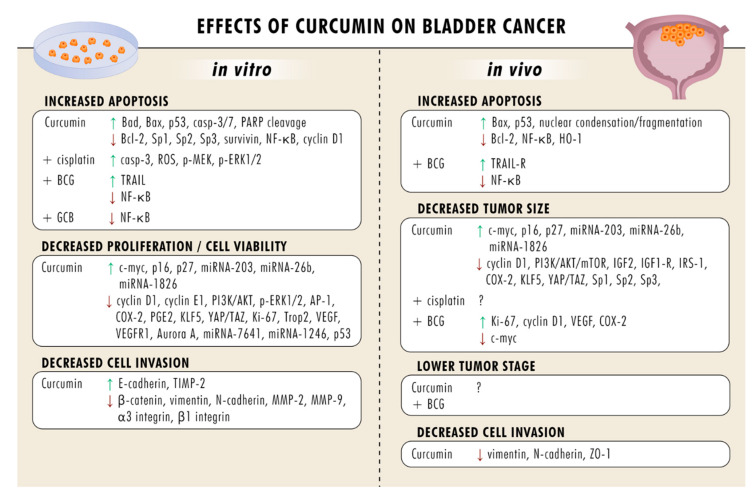
Overview of curcumin’s effects in vitro and in vivo. Abbreviations: Akt: proteinkinase B; AP-1: activator protein 1, Bad: BCL2 associated agonist of cell death; Bax: Bcl-2-associated X protein; Bcl-2: B-cell lymphoma 2; casp-3/7: caspase3/7; COX-2: cyclooxygenase-2; ERK: extracellular-regulated kinase; HO-1: heme oxygenase-1; IGF2: insulin-like growth factor 2; IGF1R: insulin-like growth factor 1 receptor; IRS-1: insulin receptor substrate 1; KLF5: Krüppel-like factor 5; miRNA: microRNA; MMP: matrix-metalloproteinase; mTOR: mechanistic target of rapamycin; NF-κB: nuclear factor ‘kappa-light-chain-enhancer’ of activated B-cells; PARP: Poly(ADP-ribose)-Polymerase 1; PGE2: prostaglandin E2; PI3K: phosphoinositide 3-kinase; ROS: reactive oxygen species; Sp 1/3/4: specificity protein 1/3/4; TAZ: transcriptional coactivator with PDZ-binding motif; TIMP-2: tissue inhibitor of metalloproteinases 2; TRAIL: tumor necrosis factor-related apoptosis-inducing ligand; Trop-2: tumor-associated calcium signal transducer 2; VEGF: vascular endothelial growth factor; VEGFR1: VEGF receptor 1; YAP: Yes-associated protein; ZO-1: Zonula occludens-1. ? = molecular background unclear.

**Table 1 ijms-21-03761-t001:** Effects of curcumin in vitro and in vivo.

Model	Cell Lines/Drugs	Outcome	Mechanism	Reference
	**COMBINATION THERAPY**			
in vitro	T24-gemcitabine resistant GCB + Curcumin	reversal of drug resistance	ABCC2; Cleaved PARP ↑ DCK; TK1;TK2; Migration ↓	[87]
in vitro	IFN-α–sensitive (RT4V6) and IFN-α–resistant (KU-7) GCB + Curcumin	increased apoptosis, IFN-α-independent	NF-κB ↓	[88]
in vitro	253J-Bv and T24 Cisplatin + Curcumin	increased apoptosis	Caspase-3; ROS ↑ p-MEK; p-ERK1/2 ↑	[90]
in vivo	nude mice, 253J-Bv xenografts Cisplatin + Curcumin	decreased tumor size	-	[90]
in vitro	253J-Bv BCG + Curcumin	increased apoptosis	TRAIL ↑; TRAIL receptor activity ↑; NF-κB ↓	[81]
in vivo	MTB-2-transplanted C3H mice BCG + Curcumin	increased apoptosis, decreased tumor size	Ki-67; CD31; NF-κB ↓; Cyclin D1; VEGF; COX-2 ↓ c-myc; Bcl-2 ↓ TRAIL receptor ↑	[81]
in vivo	F344 rats, AY27 xenografts BCG + Cyclodextrin–Curcumin	lower tumor stage	-	[80]
	**MONOTHERAPY**			
in vitro	253JB-V and KU7	increased apoptosis, cell growth inhibition	Sp1; Sp3; Sp4; Survivin ↓ VEGF; VEGFR1; p21; p27↓ Cleaved PARP ↑	[34]
in vivo	nude mice, KU7 xenografts	decreased tumor growth	Sp1; Sp3; Sp4 ↓	[34]
in vitro	5637 and WH	decreased cell viability, proliferation blockade	KLF5; YAP; TAZ; AXL ↓ ITGB2; CDK6; CYR61 ↓	[45]
in vivo	Nude mice, 5637 xenografts	decreased tumor size	YAP/TAZ; KLF5;PCNA ↓ Cyclin D1 ↓	[45]
in vitro	AY-27 (rat) and T-24,	increased apoptosis, cell cycle arrest	7-AAD; p27; Caspase-3 ↑ Cyclin D1; pRb-P; cyclin E ↓ p21; p53; NF-κB ↓	[47]
in vitro	T24	inhibited cell growth, G2/M arrest	Cyclin A; COX-2, PGE2 ↓ p21 ↑	[33]
in vitro	T24 and 5637	decreased cell growth, increased apoptosis, inhibition of migration	Caspase-3/7; TIMP-2 ↑ MMP-2; MMP-9 ↓	[53]
in vitro	T24 and 5637	proliferation blockade, increased apoptosis inhibition of migration and invasion	β-Catenin ↓ Vimentin ↓ N-cadherin ↓ E-cadherin ↑	[61]
in vitro	T24, UMUC2 and EJ	decreased cell viability, increased apoptosis, G2/M cell cycle arrest	Bcl-2; Survivin ↓ Bax; p53 ↑	[31]
in vivo	Wistar rats, N-methyl-N-nitrosourea-induced bladder cancer	increased apoptosis	Nuclear condensation and fragmentation ↑	[31]
in vitro	EJ	decreased cell viability, increased apoptosis	Intracellular esterase activity ↑ Caspase-3 ↑ DNA fragmentation ↑	[57]
in vitro	T24	decreased cell growth, G2/M cell cycle arrest	Aurora A ↓	[31]
in vitro	T24	decreased benzidine-triggered cell proliferation and G1 to S phase transition	p-ERK1/2 ↓ PCNA ↓ Cyclin D1 ↓ p21 ↑	[42]
in vitro	UMUC3 and EJ	proliferation blockade, increased apoptosis	PCNA; cyclin D1; Bcl-2 ↓ Bax; Cleaved Caspase 3 ↑ Caspase 8; Caspase 9 ↑	[56]
in vitro	5637 and BFTC 905	decreased cell viability, inhibition of invasion	MMP-2; MMP-9 ↓ ROS; HO-1 ↑	[66]
in vivo	C57BL/6 mice, MB49 xenograft		HO-1 ↑	[66]
in vitro	T24 and RT4	proliferation blockade, increased apoptosis, inhibition of mobility, G2/M cell cycle arrest	Trop2 ↓ Cyclin E1 ↓ p27 ↑	[30]
in vitro	T24 and SV-HUC-1	inhibition of invasion, increased apoptosis	miR-7641 ↓ p16 ↑	[49]
in vitro	T24 Combination with irradiation	decreased cell viability and colony formation	miR-1246 ↓	[50]
in vitro	T24	proliferation blockade, increased apoptosis	miR-203 ↑ Akt2; Src ↓	[48]
in vitro	RT112, TCCSUP and UMUC3 Combination with visible light	alteration in adhesion, inhibition of chemotaxis	RT112: pFAK; α5; β1 ↓ TCCSUP: α3; α5; β1 ↓ UMUC3: pFAK; α5; β1 ↓	[59]
in vivo	BALB/c mice exposed to tobacco smoke for 12 weeks	ameliorated EMT alterations	p-ERK1/2; p-JNK ↓ p-p38 MAPK; E-cadherin ↓ N-cadherin; ZO-1; Vimentin ↓	[62]
in vivo	C57BL/6 mice, MB49 xenograft	reduced tumor size	COX-2; Cyclin D1 ↓	[43]
in vivo	Wistar rats, N-methyl-N-nitrosourea-induced bladder cancer	decreased cell growth, inhibition of invasion	Bcl-2; Survivin ↓ Bax ↑	[55]

Abbreviations: 7-AAD: 7-Aminoactinomycin; ABCC2: ATP-binding cassette sub-family C member 2; Akt: Proteinkinase B; AXL = receptor tyrosine kinase; Bax: Bcl-2-associated X protein; Bcl-2: B-cell lymphoma 2; CD31: Cluster of Differentiation 31; CDK6: cyclin-dependent kinase 6; COX-2: cyclooxygenase-2; CYR61: cysteine-rich, angiogenic inducer, 61; DCK: deoxycytidine kinase; EMT: epithelial mesenchymal transition; ERK: extracellular-regulated kinase; HO-1: heme oxygenase-1; ITGB2: integrin beta 2; JNK: c-Jun N-terminal kinase; KLF5: Krüppel-like factor 5; MMP: Matrix-metalloproteinase; NF-κB: nuclear factor ‘kappa-light-chain-enhancer’ of activated B-cells; PARP: Poly(ADP-ribose)-Polymerase 1; pFAK: phosphorylated focal adhesion kinase; PCNA: proliferating cell nuclear antigen; PGE2: prostaglandin E2; pRb-P: phosphorylated Retinoblastom Protein; ROS: reactive oxygen species; Sp 1/3/4: specificity protein 1/3/4; SRC: Proto-oncogene tyrosine-protein kinase Src; TAZ: transcriptional coactivator with PDZ-binding motif; TIMP-2: tissue inhibitor of metalloproteinases 2; TK: thymidine kinase; TRAIL: tumor necrosis factor-related apoptosis-inducing ligand; Trop-2: tumor-associated calcium signal transducer 2; VEGF: vascular endothelial growth factor; VEGFR1: VEGF receptor 1; YAP: Yes-associated protein; ZO-1: Zonula occludens-1.

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
