# Peer review of "Curcumin—A Viable Agent for Better Bladder Cancer Treatment"

_ijms, 2020, doi:10.3390/ijms21113761_

Round 1
Reviewer 1 Report
In this manuscript, authors reviewed literature associated with the studies about the efficacy of curcumin on the molecular basis, coupled with low cytotoxicity, refining delivery procedures for the treatment of bladder cancer (BCa). Overall, this is an interesting review article and provides information regarding curcumin as a promising alternative/complementary anti-BCa agent. Several questions need to be addressed:
- Numerous recent review papers, such as J. Mol. Sci. 20(5):1033 (2019), Anticancer Agents Med Chem. 20(6):667-677 (2020), and…., have raised several important issues regarding the challenges of curcumin as a promising anticancer drug including against BCa. How the present manuscript provides novel and significantly positive inputs compared to these published article? Please highlight in the manuscript.
- Several derivatives and analogues of curcumin have been identified, developed, and well How these compounds impact on BCa compared to original curcumin?
- miRNAs have been reported to play an important role in BCa. It should add one more section to discuss the study of miRNA and curcumin in BCa.
- Any evidence showed that curcumin would be able to activate the immune response in the hosts therefore could be an attractive immune enhancer to fight cancer?
- Another question is that any publication reported that combined curcumin treatment and current immunotherapy could additively/synergistically increase the efficacy for the treatment of BCa and its aggressiveness?
- An additional figure is needed to summarize that how curcumin affects the molecular mechanism associated with cell growth/proliferation, metastasis, and apoptosis in BCa.
- What would be the potential future perspectives and works of curcumin in BCa?
- The unit: “μM” or “μmol/L” should be uniformed in the entire manuscript.
Author Response
Comment 1: Numerous recent review papers, such as J. Mol. Sci. 20(5):1033 (2019), Anticancer Agents Med Chem. 20(6):667-677 (2020), and…., have raised several important issues regarding the challenges of curcumin as a promising anticancer drug including against BCa. How the present manuscript provides novel and significantly positive inputs compared to these published article? Please highlight in the manuscript.
Our answer: Indeed, several papers are available dealing with curcumin as an anti-cancer agent. However, none of these articles focus on bladder cancer (excepting one paper, see below). Int. J. Mol. Sci. 20(5):1033 (2019) concentrates on structure activity relationships, delivery systems and effects on several cancer types (but not bladder cancer). When we started writing the review article, the Anticancer Agents Med. Chem. paper had not yet been published. Nevertheless, the anti-cancer effects of curcumin are only superficially dealt with in this paper and no information is provided with respect to tumor cell metastasis (integrin expression), chemoresistance (cisplatin, gemcitabine), immunotherapy (now included in the present article), curcumin analogues (now included in the present article), dual effects of curcumin (now included in the present article) and adverse effects (now included in the present article). We, therefore, believe that our manuscript provides a detailed overview of curcumin’s role in bladder cancer treatment, not given before.
Comment 2: Several derivatives and analogues of curcumin have been identified, developed, and well How these compounds impact on BCa compared to original curcumin?
Our answer: We only shortly dealt with curcumin derivatives, since they are not conform with the CAM concept. Still, regarding the referee’s comment, we have now included in “Curcumin delivery”: “The development of potent curcumin analogues and derivatives may also overcome limitations of the pure compound [123]. In principle, three parts of the curcumin molecule may serve as targets for structural modification: the aromatic rings, the β-diketone moiety, and the two flanking double bonds. Gao et al. recently demonstrated that pyrimidine-substituted curcumin analogues reverse multidrug resistance [124]. Deletion of the β-diketone moiety as well as mono-carbonyl curcumin analogues have been associated with superior toxicity to cancer cells, compared to the mother compound [125, 126]. Since curcumin is only poorly soluble in water, shortening of a redundant hydrocarbon chain or the introduction of hydrophilic groups offers strategies to increase its water solubility. Modifications of curcumin may include the introduction of halogens, hydroxy, nitro, and methoxy groups [127].”.
Comment 3: miRNAs have been reported to play an important role in BCa. It should add one more section to discuss the study of miRNA and curcumin in BCa.
Our answer: We have already dealt with this topic in the section “Curcumin blocks bladder cancer growth and proliferation”: miRNA-203, miRNA-26b, miRNA-1826, miRNA-7641, miRNA-1246. We have now additionally included: “Curcumin also decreases the expression of miRNA-1246, thereby increasing the expression of its target, p53, in T24 bladder cancer cells [50]. Based on a systematic review, the expression levels of miRNA-21, 143, 155, 214 and 222 have been shown to be closely associated with longer overall and progression-free survival of bladder cancer patients [51]. Although curcumin’s effects on these miRNAs have not been explored in this tumor entity, further data derived from other tumor types provide evidence that curcumin acts on these structures as well.”.
Comment 4: Any evidence showed that curcumin would be able to activate the immune response in the hosts therefore could be an attractive immune enhancer to fight cancer? Another question is that any publication reported that combined curcumin treatment and current immunotherapy could additively/synergistically increase the efficacy for the treatment of BCa and its aggressiveness?
Our answer: We have included a further section under “Curcumin triggered immune response”, which reads: “The process of tumor growth and progression is controlled by the immune microenvironment. Tumor infiltrating CD4+ and CD8+ T-cells, Treg cells, as well as innate immune cells, such as dendritic cells, macrophages and NK cells, are all involved in “immunoediting”, finally leading to an immune escape of the tumor [67]. Since this process is accompanied by increased expression of the immune checkpoint protein programmed cell death ligand 1 (PDL1) on both T-cell infiltrates and tumor cells, PDL1-inhibitors have been developed and approved for cancer treatment [68, 69]. Curcumin has also been shown to suppress PDL1 in vivo, to increase cytotoxic CD8+ T-cells and decrease Tregs and Myeloid-derived suppressor cells (MDSC) [70]. A study on patients reported that curcumin strengthens the anti-tumor immune response by converting cancer patient- derived Tregs to T helper (Th) 1 cells and enhancing interferon-gamma production [71].
Aside from regulating adaptive immunity, curcumin also holds potential to modulate innate immunity. Curcumin not only enhances the susceptibility of tumor cells towards NK mediated cell distruction [72] but also enhances the cytotoxic effect of NK cells associated with the activation of Stat4 and Stat5 proteins [73].
Studies are also underway exploring the integration of curcumin into an immunotherapeutic regimen. Curcumin has been shown to distinctly enhance adoptive T-cell therapy with improved cytotoxicity of antigen-specific CD8+ T-cells in tumor-bearing mice [74]. Combined dimethoxy derivative of curcumin together with a PDL1 antibody has been applied to metastasized bladder cancer bearing mice. Bisdemethoxycurcumin significantly increased CD8+ T-cell infiltrates and CD8+-driven interferon-gamma release and decreased the number of intratumoral MDSC. Simultaneously, the PDL1 antibody protected CD8+ T-cells from exhaustion, and therefore facilitated the secretion of interferon-gamma [75]. An elegant concept has been furnished by Hasanpoor and coworkers. In this study, a PDL1 binding peptide was used for targeted delivery of curcumin to high PDL1-expressing breast cancer cells. The strategy potently improved cellular uptake and cytotoxicity of curcumin [76]. The combination of curcumin and an anti-PDL1 antibody was found to exert synergistic anti-tumor activity in an ovarian cancer mouse model, indicating that curcumin may indeed serve as an integrative component of current immunotherapy [77]. Meanwhile, a multi-center, open-label, non-randomized, 3-cohort phase 2 study has enrolled patients with recurrent/refractory cervical carcinoma, endometrial carcinoma, or uterine sarcoma. The treatment consists of PD1 blockade (pembrolizumab) combined with radiation and food supplementation, including curcumin-intake [78]. This study is still underway.”
Comment 5: An additional figure is needed to summarize that how curcumin affects the molecular mechanism associated with cell growth/proliferation, metastasis, and apoptosis in BCa.
Our answer: We have now included an additional figure to provide the readership with an overview of curcumin’s action.
Comment 6: What would be the potential future perspectives and works of curcumin in BCa?
Our answer: We have extended the conclusion to now read: “In view of the proven manifold antitumor properties of curcumin on the molecular level, coupled with nontoxicity, this compound could serve as an augmental treatment option for bladder cancer (figure 1). Studies on humans are sparse and, therefore, ongoing trials are mandatory to evaluate whether the results obtained in vitro or in vivo will prove clinically beneficial. Curcumin dosage, bioavailability, the optimal indication, and potential toxic effects are further issues which require attention. When these issues have been clarified the question of whether curcumin can become an important new therapeutic option in bladder cancer treatment may be answered.”.
Comment 7: The unit: “μM” or “μmol/L” should be uniformed in the entire manuscript.
Our answer: This has been done.
Reviewer 2 Report
The review by Ratz et al. gives a comprehensive overview for curcumin and its application in bladder cancer.
There are several minor comments to be made.
Page 1: There are a few treatment options for metastatic bladder cancer patients with some successful applications recently. The authors may comment on that and cite a review as e.g., Nadal and Bellmunt, Cancer Treat Rev. 2019 Jun;76:10-21.
The authors may add a further article by Liang et al. Cell Death Dis., 2017, 8(10)e3066.
Page 4: The sentence: “Curcumin also decreases the expression of miRNA-1246 and its target, p53, in T24 bladder cancer cells” is somewhat misleading, since a miRNA usually represses its target gene/protein and a decrease of a miRNA results in an increase of the target gene/protein. Therefore a change to: “Curcumin also decreases the expression of miRNA-1246 and increases the expression of its target, p53, in T24 bladder cancer cells.” is suggested.
Page 5: The headline “Curcumin prevents metastatic events” is very strong and not really justified by the text. A headline as “Curcumin is suggested to prevent metastatic events” would fit the recent knowledge better.
Page 7: Add ECG (electrocardiography) to your list of abbreviations or write it out.
Although table 1 gives a detailed and comprehensive view for in vitro and in vivo studies with curcumin, an overview as figure would be preferable, e.g., as presented by Bauer and Kreis (Bauer P, Kreis W. Curcumin in der Krebstherapie Deutsche Zeitschrift für Onkologie 2011; 41: 144–149). If there are remarkable differences in vitro and in vivo, the authors may depict it in two figures.
There appeared a recent review for curcumin by Xiang et al. Medicine 2020;99:2(e18467). The authors may include it (and references therein) into their review. The figure in that article could also be of help for an overview figure (depending if the authors will focus on genes/proteins as Bauer and Kreis or on pathways as Xing et al.).
Author Response
Comment 1: Page 1: There are a few treatment options for metastatic bladder cancer patients with some successful applications recently. The authors may comment on that and cite a review as e.g., Nadal and Bellmunt, Cancer Treat Rev. 2019 Jun;76:10-21.
Our answer: Introduction now reads: “The treatment of advanced cancer includes a variety of strategies, e.g. hormonal therapy, radio-, immuno- as well as chemotherapy. The new immune checkpoint inhibitors, atezolizumab and pembrolizumab, have meanwhile widened the therapeutic options [1]”.
Comment 2: The authors may add a further article by Liang et al. Cell Death Dis., 2017, 8(10)e3066.
Our answer: We have already cited a similar article from Liang et al. (please see reference 62) but have now included the article from 2017. The section “Curcumin suppresses metastatic events” now reads: “In line with this, administration of curcumin to mice (50-100 mg/kg bw) down-regulated MAPK-signaling with subsequent EMT-alterations, i.e. elevated E-cadherin and Zonula occludens-1 (ZO-1) and reduced Vimentin and N-cadherin [62]. Suppression of the Wnt/β-catenin pathway has also been noted [63].”.
Comment 3: Page 4: The sentence: “Curcumin also decreases the expression of miRNA-1246 and its target, p53, in T24 bladder cancer cells” is somewhat misleading, since a miRNA usually represses its target gene/protein and a decrease of a miRNA results in an increase of the target gene/protein. Therefore a change to: “Curcumin also decreases the expression of miRNA-1246 and increases the expression of its target, p53, in T24 bladder cancer cells.” is suggested.
Our answer: We agree with this suggestion. The sentence now reads: “Curcumin also decreases the expression of miRNA-1246, thereby increasing the expression of its target, p53, in T24 bladder cancer cells [50]”.
Comment 4: Page 5: The headline “Curcumin prevents metastatic events” is very strong and not really justified by the text. A headline as “Curcumin is suggested to prevent metastatic events” would fit the recent knowledge better.
Our answer: We agree with this statement. To keep the headline short, it now reads: “Curcumin suppresses metastatic events”.
Comment 5: Page 7: Add ECG (electrocardiography) to your list of abbreviations or write it out.
Our answer: The respective sentence now reads: “Curcumin significantly ameliorates doxorubicin-induced electrocardiographic changes.
Comment 6: Although table 1 gives a detailed and comprehensive view for in vitro and in vivo studies with curcumin, an overview as figure would be preferable, e.g., as presented by Bauer and Kreis (Bauer P, Kreis W. Curcumin in der Krebstherapie Deutsche Zeitschrift für Onkologie 2011; 41: 144–149). If there are remarkable differences in vitro and in vivo, the authors may depict it in two figures.
Our answer: We have now included a respective figure to provide the readership with an overview of curcumin’s action.
There appeared a recent review for curcumin by Xiang et al. Medicine 2020;99:2(e18467). The authors may include it (and references therein) into their review. The figure in that article could also be of help for an overview figure (depending if the authors will focus on genes/proteins as Bauer and Kreis or on pathways as Xing et al.).
Our answer: The article of Xiang et al. primarily deals with curcumin analogues. In this context, we have extended our article in regard to chemical curcumin modifications. Xiang et al. has been referenced. The respective phrase in the section “curcumin delivery” reads: “Since curcumin is only poorly soluble in water, shortening of a redundant hydrocarbon chain or the introduction of hydrophilic groups offers strategies to increase its water solubility. Modifications of curcumin may include the introduction of halogens, hydroxy, nitro, and methoxy groups [127].”.
Reviewer 3 Report
This review article evaluates the relevance of curcumin as an integral part of therapy for bladder cancer. I like to give the following comments.
- There is no doubt regarding antitumor properties of curcumin. However, it was performed almost in vitro only. Why?
- It possessed antioxidant, anti-inflammatory, and antiapoptotic properties, meaning a multiple effect produced by curcumin. For the bladder cancer, curcumin is also effective to inhibit it through many potential mechanisms. However, the dose and/or concentration of curcumin did not conduct in the current report. Why?
- Adverse effect(s) of curcumin did not mention in this article. Additionally, dose of curcumin between effectiveness and side effect seems ignored in the discussion.
- Nontoxic is the basic requirement for agent before clinical application. However, it was not conducted in detail. Therefore, the potential of curcumin to develop as drug in clinic remained unknown.
- Metabolite(s) of curcumin were also not included in this review article.
- The main target and/or mechanism(s) of curcumin, even in cells, seems helpful to introduce in clear.
Author Response
Comment 1: There is no doubt regarding antitumor properties of curcumin. However, it was performed almost in vitro only. Why?
Our answer: The majority of curcumin related studies concentrate on molecular effects of this compound and are therefore carried out with cell culture systems. Nevertheless, in vivo studies have been published as well and cited in the present article. Human studies are sparse, which might be due to curcumin’s low bioavailability. We have already discussed this problem in the section “Curcumin delivery”. Since the synthesis of curcumin derivatives may also optimize the efficacy of curcumin, we have additionally included in “Curcumin delivery” (please see also our response to comment 4): “The development of potent curcumin analogues and derivatives may also overcome limitations of the pure compound [123]. In principle, three parts of the curcumin molecule may serve as targets for structural modification: the aromatic rings, the β-diketone moiety, and the two flanking double bonds. Gao et al. recently demonstrated that pyrimidine-substituted curcumin analogues reverse multidrug resistance [124]. Deletion of the β-diketone moiety as well as mono-carbonyl curcumin analogues have been associated with superior toxicity to cancer cells, compared to the mother compound [125, 126]. Since curcumin is only poorly soluble in water, shortening of a redundant hydrocarbon chain or the introduction of hydrophilic groups offers strategies to increase its water solubility. Modifications of curcumin may include the introduction of halogens, hydroxy, nitro, and methoxy groups [127].”.
Comment 2: It possessed antioxidant, anti-inflammatory, and antiapoptotic properties, meaning a multiple effect produced by curcumin. For the bladder cancer, curcumin is also effective to inhibit it through many potential mechanisms. However, the dose and/or concentration of curcumin did not conduct in the current report. Why?
Our answer: Due to the multiple modes of action of curcumin, we initially decided not to deal with dose-response relationships, so as not to overload the paper. Still, we have now included a short section regarding this issue (“Dose-dependent effects of curcumin”). “A publication has recently been presented pointing to dual effects of curcumin. Following curcumin treatment, the viability of renal cell carcinoma cells increased in the presence of 5 μM, did not change at 20 μM and decreased when exposed to 80 μM curcumin, compared to controls. The authors concluded that low curcumin concentrations may protect cells from autophagic cell death, whereas high concentrations may promote cell death [102]. Curcumin at 5 µM inhibited macrophage cell death by decreasing gamma-radiation induced reactive oxygen species, whereas 25 µM curcumin increased the reactive oxygen species production and augmented cell death [103]. Both anti-oxidative and pro-oxidative effects of curcumin have also been reported by others, with low dosed curcumin (1-10 µM) being a suppressor and high dosed curcumin (20 µM) being an enhancer of oxidative stress levels [104]. Curcumin’s action on angiogenic events may also depend on the drug concentration. Indeed, evidence has been provided that curcumin plays a pro-angiogenic role at 1-3 μM but an anti-angiogenic one at 10-30 μM [105]. Dose dependent effects of curcumin on blood vessel regeneration have also been demonstrated by others [106]. Whether these observations are relevant for bladder cancer treatment is still unclear. However, Alavi et al. recently showed that the combination of 5 μM curcumin with 5-fluorouracil significantly reduces the cytotoxicity of fluorouracil, while 15 μM curcumin increases cytotoxicity [107].”.
- Adverse effect(s) of curcumin did not mention in this article. Additionally, dose of curcumin between effectiveness and side effect seems ignored in the discussion. Nontoxic is the basic requirement for agent before clinical application. However, it was not conducted in detail. Therefore, the potential of curcumin to develop as drug in clinic remained unknown.
Our answer: The referee is correct. We have now included a further section: “Side effects of curcumin”, which reads: “Dose escalation studies demonstrated few curcumin side effects, when administered at high concentrations. Grade IV neutropenia and Grade III diarrhea occurred in patients (14%) with advanced or metastatic breast cancer under docetaxel when curcumin was applied orally at 8,000 mg/day for seven consecutive days [108]. Intractable abdominal fullness or pain was reported in patients (29%) with advanced pancreatic cancer under gemcitabine when the curcumin dosage was increased to 8,000 mg p.o. daily [109]. No adverse event was attributed to curcumin at 6,000 mg (6 consecutive days, p.o.) in patients with metastatic castration-resistant prostate cancer receiving docetaxel/prednisone [110]. Curcumin in complexed form with phospholipids (2,000 mg/day continuously) as a complementary approach in patients with pancreatic cancer (plus gemcitabine) did not lead to toxic events either [111]. The incidence of adverse events in cancer patients (pancreatic and biliary tract cancer) receiving gemcitabine-based chemotherapy did not increase following treatment with 400 mg of curcumin nanoparticles [89]. Short-term intravenous dosing of liposomal curcumin to healthy humans also appeared to be safe, up to a dose of 120 mg/m2. However, a transient red blood cell echinocyte formation and an increase in the mean red blood cellular volume was seen in 4% under 400 mg/m2 curcumin [112]. In a similar setting, liposomal curcumin was administered as a weekly intravenous infusion for 8 weeks in patients with metastatic tumors. No dose-limiting toxicity was observed at doses between 100 and 300 mg/m2. However, of six patients receiving 300 mg/m2, one patient developed hemolysis, and three other patients experienced hemoglobin decreases without signs of hemolysis [113].”.
- Metabolite(s) of curcumin were also not included in this review article.
Our answer: We only shortly dealt with curcumin derivatives, since they do not reflect the concept of CAM. Still, in regard to the referee’s comment, we have now included in “Curcumin delivery”: “The development of potent curcumin analogues and derivatives may also overcome limitations of the pure compound [123]. In principle, three parts of the curcumin molecule may serve as targets for structural modification: the aromatic rings, the β-diketone moiety, and the two flanking double bonds. Gao et al. recently demonstrated that pyrimidine-substituted curcumin analogues reverse multidrug resistance [124]. Deletion of the β-diketone moiety as well as mono-carbonyl curcumin analogues have been associated with superior toxicity to cancer cells, compared to the mother compound [125, 126]. Since curcumin is only poorly soluble in water, shortening of a redundant hydrocarbon chain or the introduction of hydrophilic groups offers strategies to increase its water solubility. Modifications of curcumin may include the introduction of halogens, hydroxy, nitro, and methoxy groups [127].”.
- The main target and/or mechanism(s) of curcumin, even in cells, seems helpful to introduce in clear.
Our answer: We have now added a final figure to summarize the molecular mechanism of curcumin and its effect on cell growth/proliferation, metastasis and apoptosis.

Round 2
Reviewer 1 Report
All comments have been addressed in this manuscript.